# Neonatal Respiratory Distress and Airway Emergency: Report of Two Cases

**DOI:** 10.3390/children8040255

**Published:** 2021-03-25

**Authors:** Lorenzo Bresciani, Paola Grazioli, Roberta Bosio, Gaetano Chirico, Cesare Zambelloni, Amerigo Santoro, Carla Baronchelli, Luca O. Redaelli de Zinis

**Affiliations:** 1Pediatric Otolaryngology Head Neck Surgery, Children Hospital “ASST Spedali Civili”, 25123 Brescia, Italy; lorenzobresciani87@gmail.com (L.B.); paolagrazioli85@gmail.com (P.G.); robertabosio@hotmail.it (R.B.); 2Department of Neonatology and Neonatal Intensive Care Unit, Children Hospital “ASST Spedali Civili”, 25123 Brescia, Italy; gaetano.chirico@asst-spedalicivili.it (G.C.); cesare.zambelloni@asst-spedalicivili.it (C.Z.); 3Department of Pathology, University of Brescia, 25123 Brescia, Italy; anatomia.patologica@asst.spedalicivili.brescia (A.S.); carla.baronchelli@gmail.com (C.B.); 4Department of Medical and Surgical Specialties, Radiological Sciences and Public Health, Section of Audiology, University of Brescia, 25123 Brescia, Italy

**Keywords:** congenital anomalies, intubation, tracheal agenesis, CHAOS syndrome

## Abstract

We discuss two cases of congenital airway malformations seen in our neonatal intensive care unit (NICU). The aim is to report extremely rare events characterized by immediate respiratory distress after delivery and the impossibility to ventilate and intubate the airway. The first case is a male twin born at 34 weeks by emergency caesarean section. Immediately after delivery, the newborn was cyanotic and showed severe respiratory distress. Bag-valve-mask ventilation did not relieve the respiratory distress but allowed for temporary oxygenation during subsequent unsuccessful oral-tracheal intubation (OTI) attempts. Flexible laryngoscopy revealed complete subglottic obstruction. Postmortem analysis revealed a poly-malformative syndrome, unilateral multicystic renal dysplasia with a complete subglottic diaphragm, and a tracheo-esophageal fistula (TEF). The second case is a male patient that was vaginally born at 35 weeks. Antenatally, an ultrasound (US) arose suspicion for a VACTERL association (vertebral defects, anal atresia, TEF with esophageal atresia and radial or renal dysplasia, plus cardiovascular and limb defects) and a TEF, and thus, fetal magnetic resonance (MRI) was scheduled. Spontaneous labor started shortly thereafter, before imaging could be performed. Respiratory distress, cyanosis, and absence of an audible cry was observed immediately at delivery. Attempts at OTI were unsuccessful, whereas bag-valve-mask ventilation and esophageal intubation allowed for sufficient oxygenation. An emergency tracheostomy was attempted, although no trachea could be found on cervical exploration. Postmortem analysis revealed tracheal agenesis (TA), renal dysplasia, anal atresia, and a single umbilical artery. Clinicians need to be aware of congenital airway malformations and subsequent difficulties upon endotracheal intubation and must plan for multidisciplinary management of the airway at delivery, including emergency esophageal intubation and tracheostomy.

## 1. Introduction

Congenital high airway obstruction syndrome (CHAOS) is a rare and frequently fatal disorder caused by complete or near-complete obstruction of the fetal upper airway [1]. CHAOS often results in stillbirths, while severe respiratory distress, hypoxia, absence of audible cry, and failure to intubate the airway are typical clinical features in newborn survivors, immediately after delivery. Therefore, prenatal diagnosis is desirable, even though difficult, in order to plan for ex-utero intrapartum treatment (EXIT) [2,3,4,5]. Laryngeal atresia (LA) and tracheal agenesis (TA) are two of the rare causes of intrinsic airway obstruction in CHAOS [4,5].

TA is a rare congenital airway malformation, characterized by complete or partial absence of the trachea, frequently associated with carino-esophageal fistula, broncho-esophageal fistula, or tracheo-esophageal fistula (TEF). TA was first described by Payne in 1900 and later classified into three types by Floyd, as illustrated in Figure 1 [6,7,8].

Faro et al. later proposed an alternative classification, including seven different categories, as shown in Figure 2 [8,9].

The incidence is less than 1:50,000, with a male preponderance and a high correlation with isolated or multiple congenital malformations, usually part of the “VACTERL” association (vertebral defects, anal atresia, TEF with esophageal atresia and radial or renal dysplasia, plus cardiovascular and limb defects), and the “TARCD” association (TA or tracheal atresia, radial ray defects, complex congenital cardiac abnormalities, and duodenal atresia) [3,10,11]. LA is characterized by complete or near-complete laryngeal obstruction and is classified according to either the anatomic site of obstruction or the presence of an associated TEF and the following clinical presentations, as shown in Table 1 [2,12].

LA is usually sporadic, although it can manifest as part of syndromes or chromosomal anomalies [4].

We present two cases of TA and LA observed in our NICU in a four-month period. The aim of this paper is to discuss rare congenital airway malformations which present as emergencies at delivery if not identified during pregnancy, and which carry an overall poor prognosis despite intensive treatment [10,13,14].

## 2. Case Presentation

### 2.1. Case One

A male twin was born at 34 weeks by emergency cesarean section to a 37-year-old mother. Antenatal history was notable, with in vitro fertilization, intrauterine growth retardation, and left kidney dysplasia. Birth weight was 1490 g. Immediately after delivery, bradycardia, cyanosis, and respiratory distress occurred. Apgar score was 0 and 1 at 1 and 5 min, respectively. Ventilation with a T-piece resuscitator (25/6 cm H_2_O) commenced, and FiO_2_ gradually increased to 100%. OTI with a 2.0 endotracheal tube (ETT) was attempted, but significant difficulties were encountered. Cardio-pulmonary resuscitation with continuous chest compression was performed for 1 min, and adrenaline (0.3 mg) was administered twice through the umbilical vein. Under mechanical ventilation, with 30% FiO_2_, O_2_ saturation reached 98% and heart rate reached 140 bpm. Due to persistent difficulties in ventilation and worsening combined acidosis, several attempts at OTI were performed without success. The ear-nose-throat (ENT) specialist was then asked to perform an intubation under flexible laryngoscopy guidance, although advancement of the ETT proved impossible due to an obstruction past the vocal cords. A laryngeal mask was then placed to allow for temporary ventilation. Echocardiography revealed the presence of a patent oval foramen and ductus arteriosus, severely depressed myocardial kinesis, and pulmonary hypertension. Since the newborn’s general and neurological conditions had severely deteriorated, after counselling with the parents, the decision to withdraw treatment was taken. Postmortem analysis revealed a polymalformative syndrome characterized by a subglottic complete diaphragm with a TEF 1 cm caudally to the diaphragm, as shown in Figure 3 and Figure 4, and unilateral multicystic renal dysplasia.

### 2.2. Case Two

A male was vaginally born at 35 weeks of gestation to a 43-year-old mother affected by gestational diabetes. On the first ultrasound (US) prenatal screening performed at our institution, a right multicystic kidney was documented. Subsequent US confirmed the evidence of multicystic kidney and polyhydramnios, further raising suspicion for anal atresia, TEF, and VACTERL association. Therefore, after counselling the parents regarding the possibility of a polymalformative syndrome, a fetal magnetic resonance imaging MRI was planned after six days to assess fetal airway patency. However, three days after US examination, the delivery abruptly occurred. Birth weight was 2540 g. Absence of cry, respiratory distress unresponsive to bag-valve-mask ventilation, and bradycardia occurred immediately after delivery. Intubation with a 2.5 ETT was unsuccessful, and a laryngeal mask was placed, achieving low but just sufficient blood oxygenation with FiO_2_ 100% and stabilization of heart rate. Apgar score was 2 at 1 and 5 min. Suspecting a TEF, an ETT was then placed in the esophagus, and though oxygenation remained sufficient, ventilation did not improve. The patient was transferred to the NICU, where the ENT surgeon performed a flexible laryngoscopy that documented complete sub-glottic stenosis. A laryngeal mask was placed, and mechanical ventilation was started along with administration of dobutamine, but progressive severe acidosis started to develop. Therefore, an emergency tracheostomy was attempted, but no trachea could be found except for a proximal atretic tracheal remnant. The patient expired due to worsening respiratory insufficiency and bradycardia unresponsive to cardiopulmonary resuscitation. Post-mortem analysis confirmed TA (Floyd type II/Faro type D), as illustrated in Figure 5, right kidney multicystic dysplasia, anal atresia, and a single umbilical artery consistent with the VACTERL association.

## 3. Discussion

TA and LA are rare congenital upper-airway malformations. Although sharing a common origin and few initial developmental steps, the larynx, trachea, and lungs undergo relatively separate development, allowing for malformations of the trachea in the presence of a normal larynx and vice-versa. LA is presumed to be the consequence of defective recanalization of the epithelial lamina, which in turn hampers the expansion of the subglottic region [15]. However, even the mechanism of epithelial lamina recanalization is still under investigation [16]. The trachea originates from the compartmentalization of the primitive foregut tube, set between the developing lung buds and the future larynx [11,15]. An incomplete or aberrant (ventral or dorsal) septation is thought to be responsible for TEF and tracheal or esophageal atresia/agenesis, respectively [11,17]. Even though TA is reproducible in a mouse model by inactivation of the bone morphogenic proteins in the ventral ectoderm, there is still no consensus on the precise mechanism of compartmentalization [11,18].

Prenatal US screening is the first exam that can possibly alert the clinician to a diagnosis of CHAOS [1,10]. CHAOS can be detected by US when a TEF is lacking; classical signs are enlarged hyperechogenic lungs, fluid-filled dilated trachea, and bronchi with absent flow in the trachea during breathing with or without cardiac dysfunction, diaphragmatic flattering, and massive ascites. [1,3,19,20]. However, a TEF is frequently associated with TA and LA, making US diagnosis challenging as polyhydramnios is the single most common, though non-specific, prenatal finding [3,10,19,21]. Once other causes of polyhydramnios have been excluded and fetal airway obstruction is suspected, fetal MRI can confirm the diagnosis and locate the site of obstruction, and complete genetic evaluation is essential to rule out genetic anomalies and syndromes [3,4,19,22]. In case of prenatal diagnosis, pregnancy counselling is mandatory, especially in case of genetic anomalies or near complete TA which are incompatible with survival despite treatment [22]. When pregnancy is continued, and the cause of airway obstruction can be successfully surgically managed, fetal surgery can be attempted under specific circumstances. Otherwise, an EXIT can be planned to secure the airway to allow for further evaluation and delay surgical correction of the obstruction [4,22].

Postnatal diagnosis is made by recognition of a combination of clinical signs: respiratory distress with breathing movements without appropriate air entry and absence of audible cry [2,3,10]. Preterm labor, low birth weight (<2500 g), and an Apgar score below 7 at 5 min are often noticed [1]. The newborn desaturates quickly, becoming bradycardic, while clinicians face the unanticipated impossibility of endotracheal intubation. Bag-valve-mask ventilation allows for temporary oxygenation if a TEF is present, even though ventilation remains unsatisfactory as airflow resistance is high, whereas a laryngeal mask would be ineffective unless gas leakage progresses into the esophagus [10,20,23]. However, bag-valve-mask ventilation leads to progressive gastric distention, which in turn lowers pulmonary compliance, further impairing ventilation and ultimately leading to the need for repeated gastric decompressions [23]. Once other difficult airway causes have been ruled out and more advanced intubation techniques fail, if a TA is suspected and the patient improves with bag mask ventilation, the best option to secure the airway, pending further evaluation, is esophageal intubation [10,20]. Esophageal intubation is still a temporary measure because it also causes progressive gastric distension, thus worsening respiratory distress. To secure the airways for a longer period and to allow definition of possible treatment of the malformation, different options have been described: insertion of a tracheal tube through the esophagus into the fistula [24] or positioning a temporary clip at the gastroesophageal junction by a laparotomy and a preemptive gastrostomy tube [25]. Next, an emergency cervical exploration should be attempted to perform a tracheotomy that is possible in case of LA and in patients with a Faro type E (or Floyd type 1) TA, given that the distal trachea is long enough. Nonetheless, treatment of TA and LA is always challenging as there is no clear recommendation on the best approach, and there is little time for airway assessment and intervention before severe hypoxic brain damage occurs [4,10].

After delivery, our first patient was obviously inadvertently esophageally intubated, and the CHAOS diagnosis was missed. In fact, when ventilation issues and severe combined acidosis arose, all subsequent attempts at tracheal re-intubation failed. Severe hypoxic neurological damage had already developed when the ENT specialist attempted fiber optic guided intubation and observed a subglottic LA, and after counselling with the parents, the decision to withdraw treatment was taken.

In the second case, even if prenatal US raised the suspicion of CHAOS and VACTERL association, and fetal MRI was planned, no imaging was performed and no EXIT procedure was planned due to the abruptly anticipated delivery. In this case, CHAOS was correctly diagnosed at birth and esophageal intubation was performed, but no trachea could be found on neck exploration and tracheostomy was not possible. In the end, ventilation through the carino-esophageal fistula proved insufficient, and cardiorespiratory arrest occurred due to acute severe hypoxic injury to the heart and brain.

While LA seldom allows for fetal intervention and tracheostomy at birth, TA is almost universally fatal even in highly specialized centers, and survivors are exceptional [10,22]. In a systematic review, Smith et al. reported a mortality rate of 87.2% within the first week of life and 92.6% at 1 year; 41.3% of patients did not survive beyond the first 24 h of life [10]. Those few patients who survive postnatally require the presence of a functioning TEF that allows lifesaving esophageal intubation, followed by prompt distal esophageal occlusion and gastric decompression to allow for efficient ventilation. Subsequently, distal esophagostomy and esophageal neotrachealization or reconstruction is required [3,13,24,26]. Tazuke et al., in 2015, described four patients with TA with long-term survival (77–109 months) after airway and esophageal reconstruction [13]. Densmore et al. performed esophageal trachealization and esophagocarinoplasty to treat a Floyd II TA, modifying Tazuke’s surgical approach [26]. There are a range of surgical techniques described as options to reconstruct the airway; however, the first procedure to establish an airway is always esophageal tracheostomy [10]. Nonetheless, regarding long-term prognosis of TA, the patient’s predicted survival is poor and may be further reduced by cerebral hypoxia after resuscitation [13].

## 4. Conclusions

Clinicians need to be aware of congenital airway malformations, as prenatal diagnosis is mandatory to opt for pregnancy termination or plan for a multidisciplinary management of delivery. Conversely, whenever prenatal diagnosis is impossible, difficulties upon endotracheal intubation and several clinical features should immediately alert the clinician of a possible airway malformation as there is little time for emergency management of the airway, including intubation through esophageal fistula or tracheostomy.

## Figures and Tables

**Figure 1 children-08-00255-f001:**
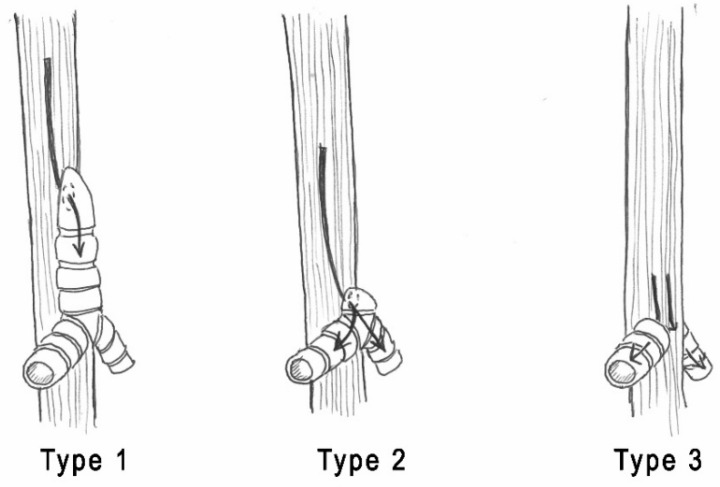
Floyd classification: (**Type 1**) The proximal trachea is atretic, the carina is normal, and the distal tracheal segment is connected to the esophagus through a fistula; (**Type 2**) the trachea is absent, and the main bronchi join at the carina. A carino-esophageal fistula is almost inevitably present; (**Type 3**) the trachea and the carina are missing, and the main bronchi directly join the esophagus.

**Figure 2 children-08-00255-f002:**
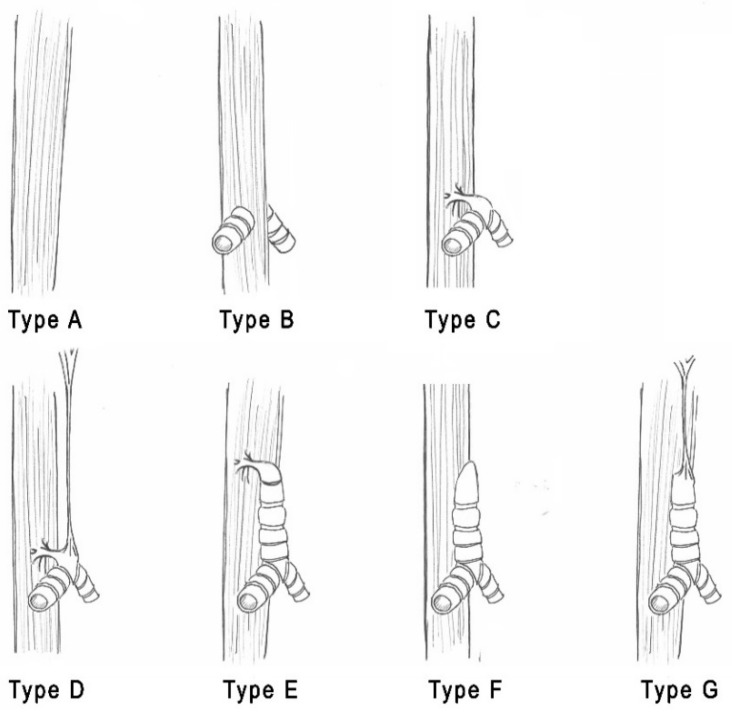
Faro classification: (**Type A**) Tracheopulmonary agenesis; (**Type B**) tracheal agenesis (TA), bronchi attached to the esophagus; (**Type C**) TA, main bronchi connected to the carina with carino-esophageal fistula; (**Type D**) TA with an atretic strand between the trachea and the larynx, carino-esophageal fistula; (**Type E**) proximal TA and distal tracheo-esophageal fistula (TEF); (**Type F**) blind bronchial bifurcation, no esophageal communication; (**Type G**) short segment TA without esophageal communication.

**Figure 3 children-08-00255-f003:**
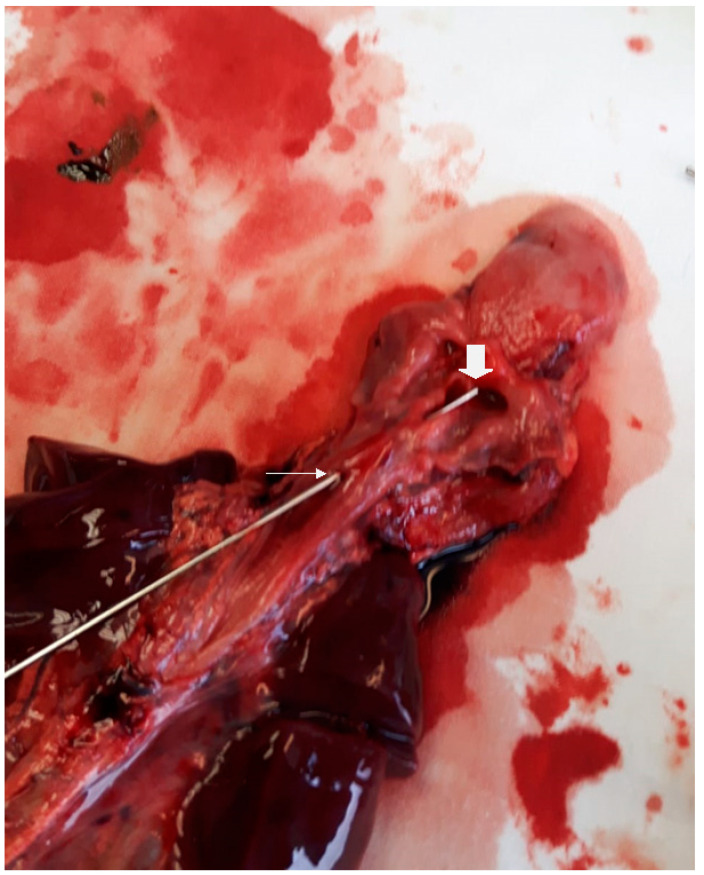
Postmortem specimen. The esophagus was opened posteriorly: a metallic probe is passed through the laryngeal lumen perforating the subglottic obstruction (**large white arrow**); the metallic probe passes through the TEF (**small white arrow**).

**Figure 4 children-08-00255-f004:**
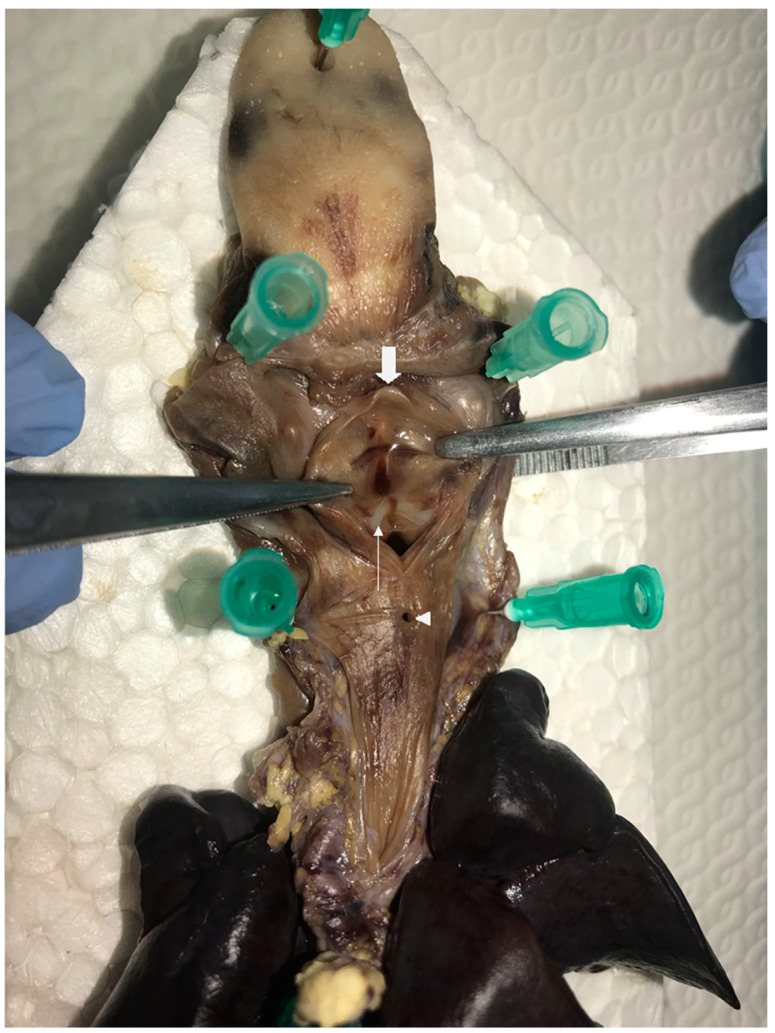
Postmortem formalin-fixed specimen. The larynx is opened posteriorly: hypoplastic epiglottic cartilage (**large white arrow**); fibro-cartilaginous subglottic diaphragm sectioned on the midline (**small white arrow**); TEF (**white triangle**).

**Figure 5 children-08-00255-f005:**
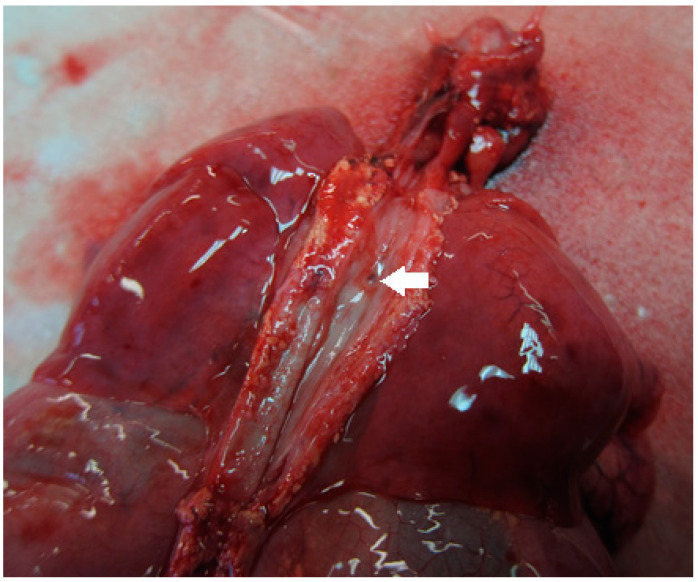
Postmortem specimen including the esophagus, lungs, and larynx. The esophagus is sectioned posteriorly on the midline: carinoesophageal fistula (**white arrow**).

**Table 1 children-08-00255-t001:** LA classifications.

Smith & Bain [12]	Hartnick et al. [2]
Type 1: complete LA with midline fusion of arytenoid cartilages and intrinsic muscles.	Type 1: complete LA without an esophageal fistula.
Type 2: subglottic obstruction where the dome-shaped cricoid cartilage obstructs the lumen.	Type 2: complete LA with a TEF.
Type 3: occlusion of anterior fibrous membranes and fusion of arytenoid cartilages at the level of the vocal process.	Type 3: near-complete high upper airway obstruction.

LA, laryngeal atresia; TEF, tracheo-esophageal fistula.

## Data Availability

The data presented in this study are available in Hospital charts of the patients

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
