# Peer review of "Neonatal Respiratory Distress and Airway Emergency: Report of Two Cases"

_children, 2021, doi:10.3390/children8040255_

Round 1
Reviewer 1 Report
The authors report on two cases with tracheal or laryngeal atresia, who died despite intensive care soon after birth.
This report is one of many others in literature. Although the documentation of the cases is thorough the information and impact of this study is poor. The authors leave the reader helpless if he or she will get in a similar situation. The significance of prenatal diagnosis is some confusing; i.e at the end of the day we don´t know if it is possible to detect the malformation before birth or not and how. And, when we know, what to do. EXIT-procedure might be helpful to gain time for establishing an airway. But, what should we do after. The studies of Tazuke et al. are not very promising when read carefully. There are no other case reports similar to them in the following years.
Author Response
The authors report on two cases with tracheal or laryngeal atresia, who died despite intensive care soon after birth.
This report is one of many others in literature. Although the documentation of the cases is thorough the information and impact of this study is poor. The authors leave the reader helpless if he or she will get in a similar situation. The significance of prenatal diagnosis is some confusing; i.e at the end of the day we don´t know if it is possible to detect the malformation before birth or not and how.
The paragraph regarding prenatal diagnosis has been reworked (line 154-160) to be more straightforward. As clearly stated (line 155), prenatal detection of CHAOS is made based on classical US findings (subsequently listed) only when a TEF is lacking, otherwise diagnosis is difficult. In fact, as reported in a review from de Groot-van der Mooren MD et al. prenatal diagnosis of CHAOS was possible in only 5 out of 49 patients because they lacked a TEF [3]. In case of a TEF, polyhydraminos is the main finding but the cause should be investigated as it is non-specific (line 159-160). Once fetal airway obstrucion is suspected, MRI is the exam of choice (sentence line 166-167 has been reworked) as further choices rely on the identification of the type and site of obstruction.
And, when we know, what to do.
The choice of treatment depends on the site and cause of obstruction. In line 169-171 it is made clear that TA is universally fatal and pregnancy termination is indicated, especially in presence of associated anomalies. Sentence at line 171 and 172 has been reworked to make it clear that further choices depend on the type and site of obstruction (laryngeal atresia, laryngeal cyst…) and the possibility to achieve successful surgical resolution of the obstruction, citations are provided.
EXIT-procedure might be helpful to gain time for establishing an airway. But, what should we do after.
The EXIT procedure is planned only once prenatal diagnosis is clearly made and an airway CAN be established. Once EXIT is performed, the airway is secured; further definitive treatment can be delayed, and choosen, once all examinations are complete. In order to clarify this concept the sentence at line 174-175 has been reworked.
The studies of Tazuke et al. are not very promising when read carefully. There are no other case reports similar to them in the following years.
TA is an almost universally fatal condition, the studies of Tazuke and the subsequent work of Densmore have been cited for completeness, to illustrate possible surgical solution to TA employed by other authors. However, it is stated quite clearly (line 77,78,171,206,207) that despite intensive treatement, TA is almost universally fatal. Furtermore, in line 169-171 it is made clear that TA is universally fatal and pregnancy termination is indicated, especially in presence of associated anomalies.
Reviewer 2 Report
I am thankful for the opportunity to review this interesting case report. The authors report on two preterm neonates with congenital high airway obstruction, both of whom died of airway management failure after postnatal resuscitation. The manuscript offers an informative review of laryngeal atresia and tracheal agenesis and the authors summarize relevant literature well. However, while the case report itself reminds practicing neonatologists about rare, yet highly challenging airway management situations, it unfortunately contributes little to the available literature on this topic. Furthermore, some aspects of the described cases would benefit from a more detailed and critical description and analysis:
1) Case One: Postnatal resuscitation is described rather cursorily. How long did the neonate receive cardiopulmonary resuscitation? As the airway could not be secured and ventilation proved insufficient, how was cardiopulmonary resuscitation delivered (chest compressions/ventilations at a ratio of 3:1 or continuous chest compressions)?
2) Case Two: Was ante partum genetic analysis considered and/or performed? How were the parents counselled, given the severe congenital malformations? Why was the planned fetal MRI, which would have been of great importance for parental counselling and planning of postnatal management, not performed earlier? Considering the suspected tracheo-esophageal fistula, was flexible laryngoscopy/bronchoscopy directly available during postnatal stabilization? An Apgar score of 2 at five minutes suggests a severely compromised cardio-respiratory condition – did the neonate require cardiopulmonary resuscitation and if yes, for how long?
Author Response
I am thankful for the opportunity to review this interesting case report. The authors report on two preterm neonates with congenital high airway obstruction, both of whom died of airway management failure after postnatal resuscitation. The manuscript offers an informative review of laryngeal atresia and tracheal agenesis and the authors summarize relevant literature well. However, while the case report itself reminds practicing neonatologists about rare, yet highly challenging airway management situations, it unfortunately contributes little to the available literature on this topic. Furthermore, some aspects of the described cases would benefit from a more detailed and critical description and analysis:
1) Case One: Postnatal resuscitation is described rather cursorily. How long did the neonate receive cardiopulmonary resuscitation? As the airway could not be secured and ventilation proved insufficient, how was cardiopulmonary resuscitation delivered (chest compressions/ventilations at a ratio of 3:1 or continuous chest compressions)?
Paragraph at line 85-90 has been extensively reworked and further details added
2) Case Two: Was ante partum genetic analysis considered and/or performed? How were the parents counselled, given the severe congenital malformations? Why was the planned fetal MRI, which would have been of great importance for parental counselling and planning of postnatal management, not performed earlier?
Paragraph at line 116-122has been reworked to clarify this fact. The patient was referred to our hospital once pregnancy was already long initiated. The first US revealed only a muticystic kidney and no further anomalies, therefore a follow-up US was planned. Once a polymalformative syndrome and TEF were suspected during the second US an MRI was immediately planned and the parents were informed, however, further pre-natal (and post-natal) counselling was scheduled after the MRI. The pre-term delivery occurred only 2 days after the US examination making impossible to proceed with imaginig, genetic analysis and complete counselling.
Considering the suspected tracheo-esophageal fistula, was flexible laryngoscopy/bronchoscopy directly available during postnatal stabilization?
- No pediatric ENT specialist is readily available, however, it was immediately requested.
An Apgar score of 2 at five minutes suggests a severely compromised cardio-respiratory condition – did the neonate require cardiopulmonary resuscitation and if yes, for how long?
Cardiopulmonary resuscitation is not reported because, despite the Apgar score, it was needed only after surgery was commenced as ventilation through the esophagus proved sufficient. However, as sugery was performed, diagnosis of TA was made clear and cardiopulmonary resuscitation proved futile since the newborn died soon after.
Round 2
Reviewer 1 Report
no comments
Author Response
We thanks the Reviewer for previous suggestions to improve the quality of the manuscript
Reviewer 2 Report
I would like to thank the authors for their responses and implemented changes to the manuscript, which helped improving the manuscript to some extent. Unfortunately, when comparing the present case report to other published literature reviews and case series/reports, it still does not provide much novel information or insight, neither from a clinical nor a pathologist’s standpoint. Smith MM et al. (Int J Pediatr Otorhinolaryngol 2017) summarized a total of 149 cases of tracheal atresia in their systematic literature review, and several other reports have been published since (e.g. S Kyle Gonzales et al. Int J Pediatr Otorhinolaryngol 2018; Lee NM et al. Medicine (Baltimore) 2019). The same is the case with laryngeal atresia, which has more recently been described by Ashraf A et al. (Case Rep Radiol 2020), Lupariello F et al. (Fetal Pediatr Pathol 2020) and El Moussaoui K et al. (Pan Afr Med J 2021), among others. Furthermore, Naina P et al. (BMJ Case Rep 2018), Saadi S et al. (Radiol Case Rep 2020) and Krishnamurthy K et al. (Fetal Pediatr Pathol 2020) have published reports on cases of tracheal agenesis in recent years.
As a side note, the clinical information provided for the postnatal course of patient 2 is somewhat contradictory. The manuscript says that “Intubation with a 2.5 ETT was unsuccessful due to subglottic obstruction, and thus a laryngeal mask was placed allowing for temporary ventilation with FiO2 100%. Apgar score was 2 at 1 and 5 minutes. An ETT was then placed in the esophagus allowing for sufficient blood oxygenation. However, ventilation remained unsatisfactory.” Yet, in their response the authors stated that “Cardiopulmonary resuscitation is not reported because, despite the Apgar score, it was needed only after surgery was commenced as ventilation through the esophagus proved sufficient.” An Apgar score of 2 at five minutes after birth suggests bradycardia and/or severely insufficient respiration, and the conflicting statements (“ventilation remained unsatisfactory” versus “ventilation through the esophagus proved sufficient”) does not help to comprehend the clinical situation.
Author Response
The revised manuscript was checked by a native English speaker 1) We are aware that description of laryngeal and tracheal atresia is not novel, but we think that the clinical aspects of these disorders, even if extremely rare, should be considered to be included in a special issue dedicated to Neonatal Respiratory Distress. Neonatologists should be aware to immediately involve otolaryngologists when respiratory distress is evident at delivery, even if prenatal diagnostic procedures failed to reveal laryngeal or tracheal obstruction as in patient 1 where a prompt tracheotomy could possibly have saved the patient’s life or when preterm delivery prevents a correct diagnosis as in patient 2. The recent case reports that you cited were published in generalist journals (Pan Afr Med J, BMJ Case Rep), in journals prevalently dedicated to otolaryngologists (Int J Pediatr Otorhinolaryngol), radiologists, (Radiol Case Rep) or pathologists (Fetal Pediatr Pathol), but we think that the different target of “Children” is more adequate to share information on these disorders.2) We rewrote the description of clinical evolution of patient 2 to clarify the steps that led to death: limited oxygenation was obtained with laryngeal mask and subsequent esophageal intubation, but ventilation was never sufficient leading to worsening respiratory insufficiency and cardiac arrest.